# PolSAR Image Classification Based on Relation Network with SWANet

**Wenqiang Hua** *⑩, **Yurong Zhang** ⑩, **Cong Zhang** ⑩ and **Xiaomin Jin** ⑩

Shaanxi Key Laboratory of Network Data Analysis and Intelligent Processing, School of Computer Science and Technology, Xi'an University of Posts and Telecommunications, Xi'an 710061, China
* Correspondence: huawenqiang@xupt.edu.cn; Tel.: +86-181-8264-4814

**Abstract:** Deep learning and convolutional neural networks (CNN) have been widely applied in polarimetric synthetic aperture radar (PolSAR) image classification, and satisfactory results have been obtained. However, there is one crucial issue that still has not been solved. These methods require abundant labeled samples and obtaining the labeled samples of PolSAR images is usually time-consuming and labor-intensive. To obtain better classification results with fewer labeled samples, a new attention-based 3D residual relation network (3D-ARRN) is proposed for PolSAR image. Firstly, a multilayer CNN with residual structure is used to extract depth polarimetric features. Secondly, to extract more important feature information and improve the classification results, a spatial weighted attention network (SWANet) is introduced to concentrate the feature information, which is more favorable for a classification task. Then, the features of training and test samples are integrated and CNN is utilized to compute the score of similarity between training and test samples. Finally, the similarity score is used to determine the category of test samples. Studies on four different PolSAR datasets illustrate that the proposed 3D-ARRN model can achieve higher classification results than other comparison methods with few labeled data.

**Keywords:** PolSAR image; terrain classification; relation network; attention

## 1. Introduction

Polarimetric synthetic aperture radar (PolSAR) is an active microwave imaging technology. Compared to traditional SAR technology, PolSAR adopts the multi-channel and multi-polarization working mode, which can obtain rich target information through the transmission and reception of polarimetric electromagnetic waves. Because of these advantages, PolSAR image has made outstanding achievements in remote sensing applications in a variety of fields [1–3]. PolSAR image terrain classification is a very important fundamental project in these applications, which aims to classify the pixels of the whole map into the corresponding categories through the polarimetric information. With the continuous evolution of machine learning technology, various PolSAR classification methods have been presented [4–6]. Based on the availability of labeled samples, these methods can be classified as unsupervised, semi-supervised, and supervised methods.

In general, unsupervised classification methods of PolSAR images do not require labeled samples. They mainly study the polarimetric scattering mechanism [7–12] and statistical distributions of polarimetric features [13], such as Pauli decomposition, Freeman three-component decomposition, Cloude decomposition, Huynen decomposition, and Krogager decomposition. In addition, researchers also studied other polarimetric features, such as the roll-invariant features [14]. Based on these polarimetric features, many unsupervised classification methods were proposed to improve the classification accuracy. Although the unsupervised method is simple and fast, the classification accuracy is not good due to the lack of samples with labels. Unlike the unsupervised classification, the supervised method obtains better classification performance by utilizing labeled samples. For PolSAR images,

some representative supervised methods have been proposed, such as k-nearest neighbor (KNN) [15], support vector machine (SVM) [16], and neural networks [17].

Recently years, deep learning, as a main neural network method, has achieved remarkable results on many problems, such as object detection [18], image classification [19], and target recognition [20]. With the continuous progress of deep learning, deep learning-based classification algorithms have been proposed for PolSAR images, such as the deep belief network (DBN) [21], the stacked autoencoder (SAE) [22], and convolutional neural networks (CNN) [23]. However, the classification accuracy of PolSAR images is significantly enhanced by spatial information [22]. CNN can well obtain the spatial feature of an image with its unique network structure. Shang et al. [24] proposed a spatial feature-based convolutional neural network (SF-CNN) to solve PolSAR classification problem in a limited labeled dataset. Cui et al. [25] optimized topologies of basic CNNs by adding multipath. Therefore, these CNN-based PolSAR classification methods are proposed to achieve higher classification precision. However, deep learning methods require sufficient labeled samples and collecting labeled samples usually requires a lot of laborers and materials for PolSAR images.

In contrast, it is easier to obtain unlabeled samples than labeled samples, which can provide more statistical information to compensate for the limited labeled samples. This fact promoted the concept of semi-supervised learning, which enables training with both labeled and unlabeled samples to obtain potential performance improvement. Therefore, some semi-supervised methods [26,27] have been put forth to figure out the issue of PolSAR image classification with limited data, such as the co-training method, tri-training method, self-training method, and graph-based method. Recently, in order to use the capabilities of deep learning to tackle the categorization challenge of PolSAR images with limited data, researchers proposed several semi-supervised methods based on deep learning. Fang et al. [28] proposed a semi-supervised 3D-CNN model using pseudo labels. Guo et al. [29] combined a memory mechanism and a semi-supervised learning method to construct a semi-supervised method based on a memory convolutional network for PolSAR classification. Although these methods have made great advances, it is still challenging to classify PolSAR images with deep learning methods especially for the limited labeled samples. Accordingly, this paper mainly studies the classification of PolSAR images with only a few labeled samples.

The main purpose of few-shot learning is to identify new categories from a small number of labeled data [30]. Although, the availability of few labeled samples hampers the typical deep learning fine-tuning process, some methods have made considerable advances in few-shot learning, such as matching networks [31], prototypical networks [32], and meta-learning [33]. The matching network identifies the unlabeled data (query set) using attention embedding matching measurements between labeled data (support set). The prototype network learns metric space and realizes classification by calculating the distance to the prototypical representation of each class. Meta-learning attempts to acquire a collection of projection functions so that when an image is represented in such an embedding, the image can be easily identified using a simple nearest neighbor or linear classifier.

In recent years, few-shot learning has attracted numerous interests in the subject of remote sensing. Liu et al. [34] suggested a residual network-based few-shot learning method for hyperspectral image classification that learns a metric space based on episode training samples of the source domain, then combines it with a basic classifier to improve the classification results. Gao et al. [35] developed a related network for hyperspectral image classification with few labeled samples. Tong et al. [36] introduced a few-shot hyperspectral image classification method based on attention-weighted graph convolutional networks, which use the GNN to few-shot hyperspectral image classification. Zuo et al. [37] suggested a hyperspectral image classification algorithm based on edge-labeling graph neural networks, which use a graph neural network to explicitly quantify the relationships between pixels. Zhang et al. [38] proposed a few-shot unsupervised deep representation

learning based on contrastive learning for PolSAR classification, which learns transferrable representations from unlabeled data.

Although few-shot learning has achieved remarkable achievements in other remote sensing image classifications, the application in PolSAR images is still less common. Relation networks (RN) [30] is one of the few-shot learning methods based on meta-learning-based models, which is simple, flexible, efficient, and has obtained some excellent performances in image recognition [35,39]. However, the RN was initially designed for a natural image rather than a PolSAR image without considering the difference between them. For a natural image, an image is a sample, while a pixel is a sample in a PolSAR image. For a PolSAR image, each pixel has its spatial neighborhood, which is very important in deciding the category of the pixel, but natural images have no such neighborhood. Moreover, compared with natural images, PolSAR images have a more complex intraclass similarity and interclass difference, which requires more sufficient and valid features.

Based on the above analysis, this paper proposes a new few-shot learning classification method for PolSAR images based on RN. However, it should be noted that an RN requires a sufficient number of labeled samples from the source domain for training. For PolSAR images, it is very difficult to gather a sufficient number of labeled data with the same band and imaging system for training relational networks. Therefore, we proposed a training set construction method using few labeled samples to train the RN model based on the superpixels algorithm [40].

In addition, to fully extract spatial information and channel information for a PolSAR image, 3D-CNN [41,42] is used for network training and classification in the presented method. Moreover, the depth features extracted by the deep network inevitably have redundant information, and the traditional feature selection approaches are unaffected by classification. Accordingly, a spatial weighted attention network (SWANet) is proposed to automatically obtain the importance of spatial features through learning, avoiding the complex feature selection process. Then, according to the value of features, the useful features are promoted and the less useful ones are suppressed to improve classification accuracy.

Therefore, the important contributions of this study are stated below:

1. A new deep network 3D-ARRN is proposed for few-shot PolSAR image classification. This method can automatically select and extract features to achieve end-to-end final classification and effectively improve the classification results with fewer labeled samples.
2. According to the properties of a PolSAR image, a spatial weighted attention network (SWANet) is proposed to select important spatial features to improve the network performance.
3. We proposed a superpixels-based pseudo-labeled sample generation method, and used the pseudo-labeled sample to learn the transferrable representations. Then, the collected representations are transferred with limited labeled data to perform the few-shot PolSAR classification.

The remainder of this article is structured as follows. The specifics of the proposed method are presented in Section 2. Section 3 depicts the experimental design using four PolSAR data. The final experimental results are presented in Section 4. Section 5 concludes the paper.

## 2. The Proposed Method

### 2.1. Polarimetric Representation

Each resolution cell in PolSAR data is represented as a complicated coherency matrix $T$, which reflects the transformation relation between incident wave and scattered wave of target information. Its form is defined as:

$$T = \begin{bmatrix} T_{11} & T_{12} & T_{13} \\ T_{21} & T_{22} & T_{23} \\ T_{31} & T_{32} & T_{33} \end{bmatrix} \tag{1}$$

The coherency matrix $T$ is a complex conjugate symmetric matrix. Its primary diagonal elements are real numbers and the off-diagonal elements are complex numbers that satisfy conjugation. In general, the coherency matrix $T$ can be used to generate a true 6-dimensional feature vector, which is defined as:

$$
\begin{aligned}
A_T &= 10\log_{10}(T_{11} + T_{22} + T_{33}) \\
B_T &= T_{22}/(T_{11} + T_{22} + T_{33}) \\
C_T &= T_{33}/(T_{11} + T_{22} + T_{33}) \\
D &= |T_{12}|/\sqrt{T_{11} \cdot T_{22}} \\
E &= |T_{13}|/\sqrt{T_{11} \cdot T_{33}} \\
F &= |T_{23}|/\sqrt{T_{33} \cdot T_{22}}
\end{aligned}
\tag{2}
$$

where $A_T$ indicates the total scattering power. $B_T$ and $C_T$ represent the normalized power ratios of $T_{22}$ and $T_{33}$, respectively. $D_T$, $E_T$, and $F_T$ denote the relative correlation coefficients.

Apart from the coherency matrix, other scattering features needed to be involved. In our study, scattering features obtained from target decomposition methods and roll-invariant polarimetric features are further used, as list in Table 1, because different features can reflect specifics of the scattering mechanism details from various perspectives. Therefore, we combine together all these polarimetric features to construct a 32-dimensional feature in this study.

**Table 1.** PolSAR features used in this study.

| Feature | Description |
| --- | --- |
| $H, a, A, \lambda_1, \lambda_2, \lambda_3$ | Cloude decomposition [9] |
| $P_s, P_d, P_v$ | Freeman–Durden decomposition [10] |
| $|k_s|^2, |k_v|^2, |k_d|^2$ | Krogager decomposition [12] |
| $\langle A_0 \rangle, \langle B + B_0 \rangle, \langle B - B_0 \rangle, \langle C \rangle, \langle D \rangle, \langle E \rangle, \langle F \rangle, \langle G \rangle, \langle H \rangle$ | Huynen decomposition [11] |
| $|a|^2, |b|^2, |c|^2$ | Pauli decomposition [8] |
| $\theta_{null}\_\mathrm{Re}[T_{12}], \theta_{null}\_\mathrm{Im}[T_{12}]$ | Roll-Invariant Polarimetric Features [7] |
| $A_T, B_T, C_T, D_T, E_T, F_T$ | Elements from the coherency matrix |

### 2.2. Feature Selection Based on SWANet

In this paper, the 32-dimensional feature vector are extracted by coherence matrix $T$, which is used as the feature information of each pixel. After a multi-layer convolutional network, multi-dimensional deep features are extracted from 32-dimensional feature information. However, CNN-based classification methods of PolSAR images usually select pixel blocks to represent the features of central pixel points. Although these methods consider the spatial neighborhood information between pixels, the direct use of pixel blocks may affect the judgment of the center pixel category because the neighborhood pixels and the center pixels in the irregular area may belong to different categories. Therefore, to reduce the impact of different categories of pixels in the same neighborhood on the determination of the category of central pixels, this paper proposes a spatial weighted attention network (SWANet) to calculate the contribution of the neighborhood pixels in the pixel block to the center pixel. Then, the original input features are weighted by the weight after learning and the importance of the original features in the spatial range are re-calibrated. The specific process is shown in Figure 1.

As shown in Figure 1, the input feature information can be represented as a tensor $h \times w \times c$, where $h$ represents the height, $w$ denotes the width, and $c$ represents the dimension of the input feature. First, compute the spatial similarity between the central pixel and other pixels in the pixel block:

$$
d_i = \sqrt{\sum_{i=1}^{c} (x_i - x_j)^2}
\tag{3}
$$

where $x_i$ represents pixel feature information in the pixel block and $x_j$ represents the feature information of the center pixel in the pixel block.

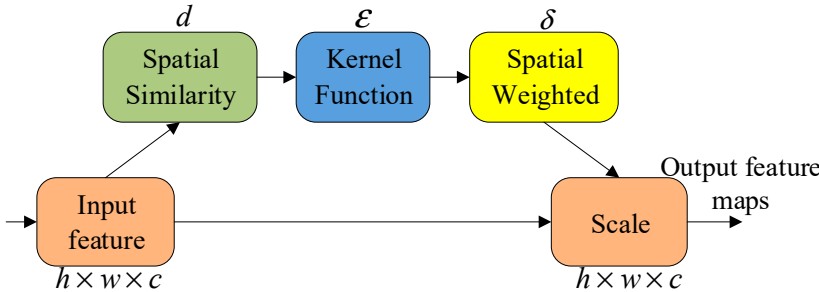

**Figure 1.** The structure of the spatial weighted attention network.

The larger the value of $d_i$, the greater the difference between the neighborhood pixels $x_i$ and center pixel $x_j$ inside the pixel block. In this case, the feature contribution of $x_i$ to the center pixel $x_j$ should be smaller when judging the category of the center pixel. Therefore, a kernel function is introduced as follows:

$$\varepsilon_i = \exp(-d_i) \tag{4}$$

Then, the weights of all the points in the pixel block are normalized as follows:

$$\delta_i = \frac{\varepsilon_i}{\sum\limits_{i=1}^{h \times w} \varepsilon_i} \tag{5}$$

Finally, the scaling operation multiplies the input information $\Lambda$ by weight parameter $W$ and outputs a new feature map $U$, $U \in R^{h \times w \times c}$,

$$U = F_{scale}(\Lambda, W) = \Lambda \cdot W \tag{6}$$

where $W$ represents the matrix of $h$ row and $w$ column composed of weights $\delta_i$.

### 2.3. Relation Network

The relation network is a classical method for few-shot learning, which is a simpler and efficient classification method and has made outstanding achievements in remote sensing applications in a variety of fields. The design idea of an RN is to constrain the functional structure of a neural network to capture the common characteristics of relational reasoning. That is to say, the ability to calculate relationships is implanted into the RN model without learning just as the ability to reason about space and translation invariant attributes is built-in in CNN.

In RN methods, there are two datasets: target domain data and source domain data. The target domain data is divided into the support set $D_t^{support}$ and query set $D_t^{query}$, and the source domain data is also called the training set $D_s^{train}$. In the $M$-way $K$-shot classification task, the support set $D_t^{support}$ contains $I = M \times K$ samples of $M$ classes with $K$ samples for each class. An RN utilizes training sets through episode-based learning. An episode consists of a randomly selected query sample $D_s^{query}$ and several support samples $D_s^{support}$ in source domain. After training, the trained parameters are stored in the RN model by executing the source domain. Then, this trained model is used to perform a new classification task. The labeled target data (support set $D_t^{support}$) and trained model are intended to predict the labels of all unlabeled target class data (query set $D_t^{query}$).

As depicted in Figure 2, the RN model contains two parts: the embedding module and the relation module. For each episode, randomly select class $M$ from the source domain and randomly select $K$ samples from each class $M$ as the sample set to form a support set

$D_s^{support} = \{(x_i, y_i)\}_{i=1}^m$ ($m = M \times K$) and select a part of these class $M$ samples as the query set $D_s^{query} = \{(x_j, y_j)\}_{j=1}^n$. For each sample pair $x_i$ and $x_j$, they are first fed into the embedding module $f_\phi$ to obtain deep features $f_\phi(x_i)$ and $f_\phi(x_j)$, respectively. Then, these feature maps ($f_\phi(x_i)$ and $f_\phi(x_j)$) are combined together with operator $C(f_\phi(x_i), f_\phi(x_j))$, where $C$ is a concatenation of feature maps in depth. Finally, the combined feature maps are input into the relation module so that the relation score $r_{i,j}$ between the query sample $x_j$ and support sample $x_i$ can be generated to indicate their similarity. The value of relation score $r_{i,j}$ is 0 to 1, and the higher value indicates a greater similarity. It is defined as:

$$r_{i,j} = g_\varphi(C(f_\varphi(x_i), f_\varphi(x_j))) \quad i = 1, 2, \ldots, m \tag{7}$$

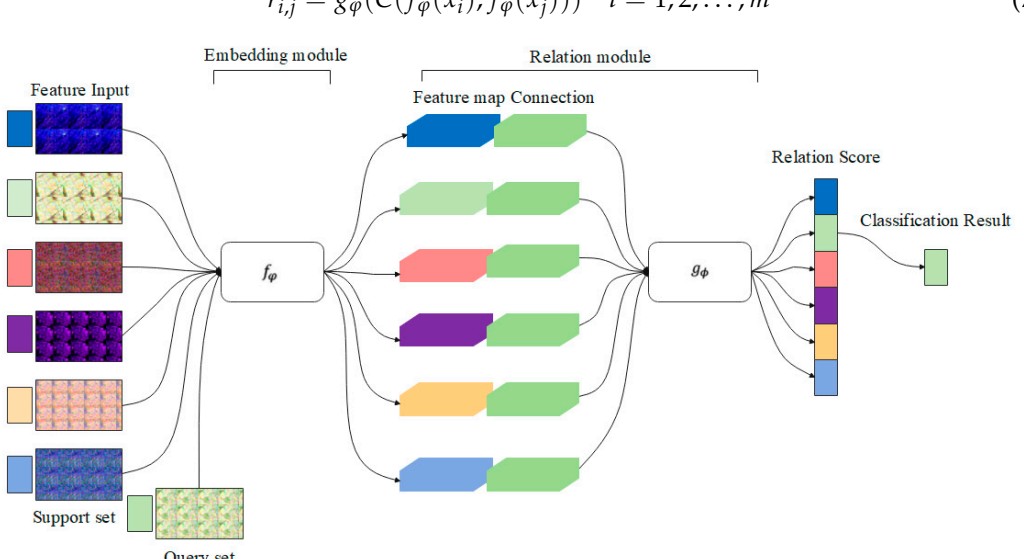

**Figure 2.** The structure of relation network.

Considering that similarity prediction is a regression problem in the RN model. In the training process, the RN model uses mean square error loss for network training, and the object function of an RN is defined as:

$$\varphi, \phi \leftarrow \underset{\varphi, \phi}{\operatorname{argmin}} \sum_{j=1}^Q \sum_{i=1}^M (r_{i,j} - 1(y_i = y_j))^2 \tag{8}$$

where $Q$ represents the quantity of the query sample, $y_i$ denotes the label for the support sample, and $y_j$ represents the label for the query sample.

*2.4. Pseudo-Labels Generation Algorithm*

In few-sample problems, the classifier trained only with support set samples has poor classification performance due to a scarcity of labeled samples. According to this problem, the RN carries out meta-learning algorithm on the source domain in order to derive transferable information so that we can carry out better feed efficiency from shot learning of the support set and more successfully classify the test set. Therefore, source domain data are essential to an RN. However, for PolSAR data, it is difficult to collect a relevant labeled sample with the same band and imaging system. As a result, a superpixel-based training set construction method is proposed that uses both limited labeled samples and significant unlabeled samples to generate a source domain data with a large number of pseudo-labels. Figure 3 shows the flowchart of this process, and the following are the specific steps:

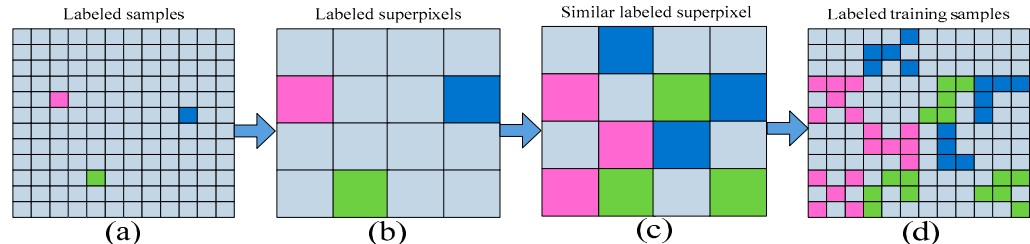

**Figure 3.** The flowchart of constructing training samples. (**a**) denotes the labeled samples; (**b**) denotes the labeled superpixels; (**c**) denotes the similar labeled superpixels; (**d**) denotes the labeled training samples.

1. Divide the Pauli image of PolSAR into $N_s$ superpixels by turbopixels algorithm [42], and calculate the clustering center of each superpixel.
2. Select the superpixel containing the support sample and give it the same label as the support sample (Figure 3b). Then, calculate the clustering center of these labeled superpixels.
3. Among all superpixels, select the superpixel most similar to each class of labeled superpixel by Equation (9) and mark it.

$$d_{i,j}(V_i, V_j) = (N_i + N_j) \ln \left| \widetilde{V} \right| - N_i \ln \left| \widetilde{V}_i \right| - N_j \ln \left| \widetilde{V}_j \right| \tag{9}$$

$$\widetilde{V}_i = 1/N_i \sum_{t=1}^{N_i} T_t \tag{10}$$

$$\widetilde{V}_j = 1/N_j \sum_{t=1}^{N_j} T_t \tag{11}$$

$$\widetilde{V} = 1/(N_i + N_j) \sum_{t=1}^{N_i+N_j} T_t \tag{12}$$

where $N$ represents the number of pixels in the superpixels. $T_t$ represents the polarimetric coherence matrix. $d_{i,j}$ denotes the similarity between $i$-th and $j$-th superpixels; the smaller value of $d_{i,j}$, the $i$-th and $j$-th superpixels are more similar.

4. Repeat step 3 until sufficient labeled superpixels are obtained (Figure 3c).
5. Calculate the distance from all pixels in each labeled superpixel to the center of the superpixel by Equation (13), and calculate the average value of these distances by Equation (14).

$$w_t(T_t, V_s) = \ln \left| V_i \right| + Tr(V_s^{-1} T_t) \tag{13}$$

$$w_c = \frac{1}{N_s} \sum_{t=1}^{N_s} w_t \tag{14}$$

where $V_s = 1/N_s \sum_{t=1}^{N_s} T_t$ denotes the center of superpixel.

6. If the distance from any pixel in the labeled pixel to the superpixel center is less than the average distance $w_c$, the label of the pixel is reserved; otherwise, the label of the pixel is removed (Figure 3d).
7. Output these pseudo-labels sample as the training data.

### 2.5. Architecture of the 3D-ARRN

In this section, a novel few-shot PolSAR image classification method is proposed based on 3D-ARRN. As depicted in Figure 4, the structure of the proposed method mainly consists of four modules: the data input module, the embedding module, the relation module, and

the final image classification. For the M-way K-shot classification problem of the PolSAR image, the proposed method first extracts polarimetric features from the coherency matrix *T* to represent the support samples and query samples. Secondly, construct a 3D embedding module based on the residual network and SWANet for feature extraction and feature selection. Then, use the 3D residual relation module to calculate the similarity of the query sample and support sample, and then calculate the objective function according to this similarity. Next, the objective function is minimized to train the proposed model in each episode. Finally, the category of the test sample is determined by relational score. The structure of the suggested method for training is summarized in Algorithm 1.

---

**Algorithm 1.** Framework of the 3D-ARRN for a training episode

---

**Input:** Support set $D_s^{support} = \{(x_i, y_i)\}_{i=1}^{m} (m = M \times K)$, query set $D_s^{query} = \{(x_j, y_j)\}_{j=1}^{n}$.

**Output:** The objective function of the RN in an episode.

1: Obtain the vector representation of each support samples and query samples.

2: The residual network and SWANet are used for feature extraction and selection of the support set and query set, and extracted features are expressed as $f_\phi(x_i)$ and $f_\phi(x_j)$, respectively.

3: Connect the feature maps of support set and query set by operator $C(f_\varphi(x_i), f_\varphi(x_j))$.

4: The combined features are fed into the relational network $g_\phi$ to calculate the relational score $r_{i,j} = g_\varphi(C(f_\varphi(x_i), f_\varphi(x_j)))$ between the query sample and the support samples and calculate the objective function in Equation (8).

5: Return the objective function to be minimized to train the model.

---

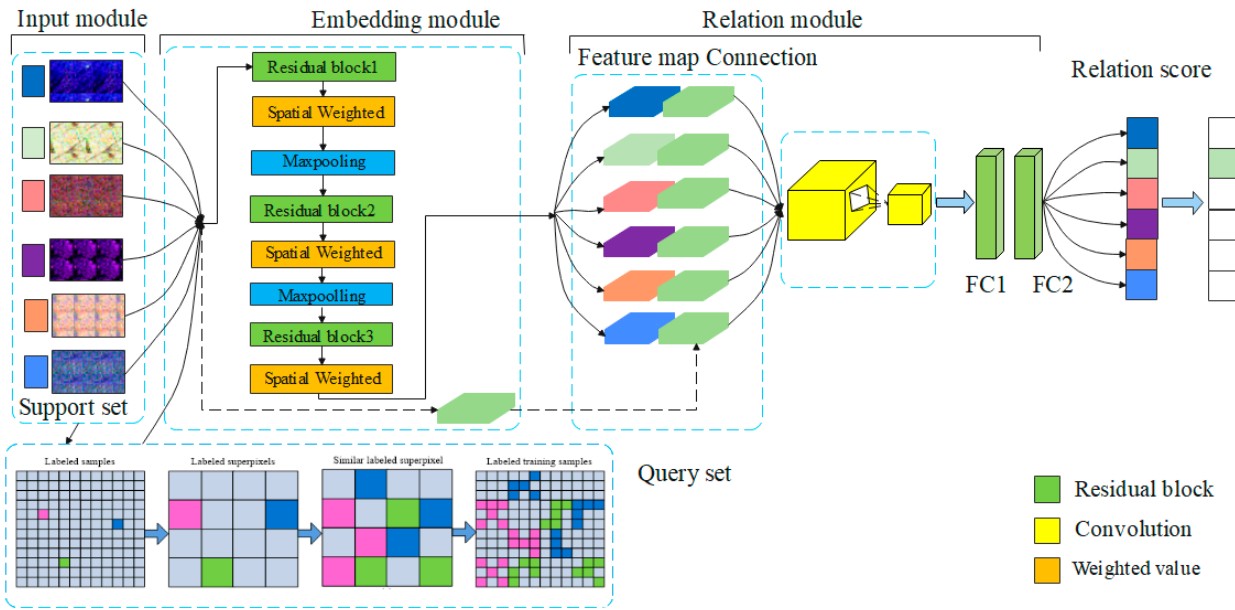

**Figure 4.** The overall network structure of the 3D-ARRN.

## 3. Experimental Design

In this section, four real PolSAR datasets are used to verify our proposed method in this paper, and all experiments were conducted on a computer with 2080Ti GPUs. The parameters are set as follows: adaptive moment estimation (Adam) is used for optimization, the learning rate has been set at 0.001, and the filter method adopts the Lee filter [43]. The specific parameters of the proposed classification network are shown in Table 2. In addition, the overall classification accuracy (OA) is taken to verify the effectiveness of our proposed method.

**Table 2.** The specific parameters of the proposed classification network.

| | Proposed Method | | | | | |
|---|---|---|---|---|---|---|
| | **Embedding Module** | | | | | |
| **Layer Name** | **Support Samples** | | | **Query Samples** | | |
| | **Output Shape** | **Filter Size** | **Padding** | **Output Shape** | **Filter Size** | **Padding** |
| Input | $K \times 1 \times 9 \times 15 \times 15$ | N/A | N | $Q \times 1 \times 9 \times 15 \times 15$ | N/A | N |
| Convolution1 | $K \times 9 \times 9 \times 15 \times 15$ | $1 \times 9 \times 3 \times 3 \times 3$ | Y | $Q \times 9 \times 9 \times 15 \times 15$ | $1 \times 9 \times 3 \times 3 \times 3$ | Y |
| Convolution2 | $K \times 9 \times 9 \times 15 \times 15$ | $9 \times 9 \times 3 \times 3 \times 3$ | Y | $Q \times 9 \times 9 \times 15 \times 15$ | $9 \times 9 \times 3 \times 3 \times 3$ | Y |
| Convolution3 | $K \times 9 \times 9 \times 15 \times 15$ | $9 \times 9 \times 3 \times 3 \times 3$ | Y | $Q \times 9 \times 9 \times 15 \times 15$ | $9 \times 9 \times 3 \times 3 \times 3$ | Y |
| Shortcut | Convolution1 + Convolution3 | | | Convolution1 + Convolution3 | | |
| Spatial Weight | $K \times 9 \times 9 \times 15 \times 15$ | N/A | N | $Q \times 9 \times 9 \times 15 \times 15$ | N/A | N |
| Max pooling1 | $K \times 9 \times 9 \times 7 \times 7$ | $1 \times 2 \times 2$ | N | $Q \times 9 \times 9 \times 7 \times 7$ | $1 \times 2 \times 2$ | N |
| Convolution4 | $K \times 32 \times 9 \times 7 \times 7$ | $9 \times 32 \times 3 \times 3 \times 3$ | Y | $Q \times 32 \times 9 \times 7 \times 7$ | $9 \times 32 \times 3 \times 3 \times 3$ | Y |
| Convolution5 | $K \times 32 \times 9 \times 7 \times 7$ | $9 \times 32 \times 3 \times 3 \times 3$ | Y | $Q \times 32 \times 9 \times 7 \times 7$ | $9 \times 32 \times 3 \times 3 \times 3$ | Y |
| Convolution6 | $K \times 32 \times 9 \times 7 \times 7$ | $9 \times 32 \times 3 \times 3 \times 3$ | Y | $Q \times 32 \times 9 \times 7 \times 7$ | $9 \times 32 \times 3 \times 3 \times 3$ | Y |
| Short cut | Convolution4 + Convolution6 | | | Convolution4 + Convolution6 | | |
| Spatial Weight | $K \times 32 \times 9 \times 7 \times 7$ | N/A | N | $Q \times 32 \times 9 \times 7 \times 7$ | N/A | N |
| Max pooling2 | $K \times 32 \times 9 \times 3 \times 3$ | $1 \times 2 \times 2$ | N | $Q \times 32 \times 9 \times 3 \times 3$ | $1 \times 2 \times 2$ | N |
| Convolution7 | $K \times 64 \times 9 \times 3 \times 3$ | $32 \times 64 \times 3 \times 3 \times 3$ | Y | $Q \times 64 \times 9 \times 3 \times 3$ | $32 \times 64 \times 3 \times 3 \times 3$ | Y |
| Convolution8 | $K \times 64 \times 9 \times 3 \times 3$ | $32 \times 64 \times 3 \times 3 \times 3$ | Y | $Q \times 64 \times 9 \times 3 \times 3$ | $32 \times 64 \times 3 \times 3 \times 3$ | Y |
| Convolution9 | $K \times 64 \times 9 \times 3 \times 3$ | $32 \times 64 \times 3 \times 3 \times 3$ | Y | $Q \times 64 \times 9 \times 3 \times 3$ | $32 \times 64 \times 3 \times 3 \times 3$ | Y |
| Shortcut | Convolution7 + Convolution9 | | | Convolution7 + Convolution9 | | |
| Spatial Weight | $K \times 64 \times 9 \times 3 \times 3$ | N/A | N | $Q \times 64 \times 9 \times 3 \times 3$ | N/A | N |
| | **Relation module** | | | | | |
| | Output Shape | | Filter Size | | Padding | |
| Convolution10 | $N_m \times 128 \times 9 \times 3 \times 3$ | | $128 \times 64 \times 1 \times 3 \times 3$ | | Y | |
| Max pooling3 | $N_m \times 64 \times 11 \times 1 \times 1$ | | $1 \times 2 \times 2$ | | N | |
| Convolution11 | $N_m \times 64 \times 13 \times 1 \times 1$ | | $64 \times 64 \times 1 \times 3 \times 3$ | | Y | |
| Flatten | $N_m \times 832$ | | N/A | | N | |
| Flatten | $N_m \times 8$ | | N/A | | N | |
| Output | $N_m \times 1$ | | N/A | | N | |

Note: $K$ and $Q$ denote the number of support samples and query samples respectively; $N_m = Q \times M$, where $M$ represents the number of categories of the classification.

### 3.1. Data Sets

The first PolSAR dataset is L-band PolSAR data of Flevoland I, the Netherlands, acquired by the AIRSAR system in 1989, which has $750 \times 1024$ pixels. The resolution of this image is $6.6m \times 12.1m$. It contains fifteen different types of crops and each type of crop is identified by one color. Figure 5a is the Pauli RGB map of this image and Figure 5b is the corresponding ground-truth map of this image.

The second PolSAR dataset is C-band PolSAR data of Flevoland II, the Netherlands, acquired by RADARSAT-2 in 2008, which has $1400 \times 1200$ pixels. The resolution of this image is $10m \times 5m$. It contains four different terrain types. The Pauli RGB map of the image is shown in Figure 6a and the corresponding ground-truth map of this image is depicted in Figure 6b.

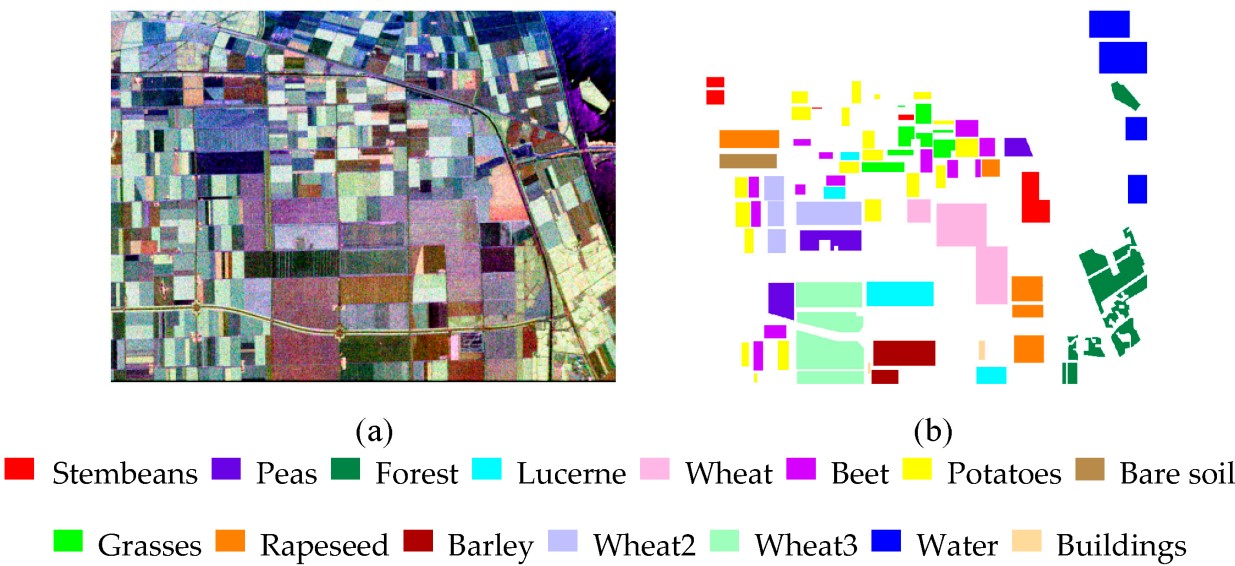

**Figure 5.** AIRSAR dataset in Flevoland I: (**a**) Pauli image. (**b**) Ground-truth image.

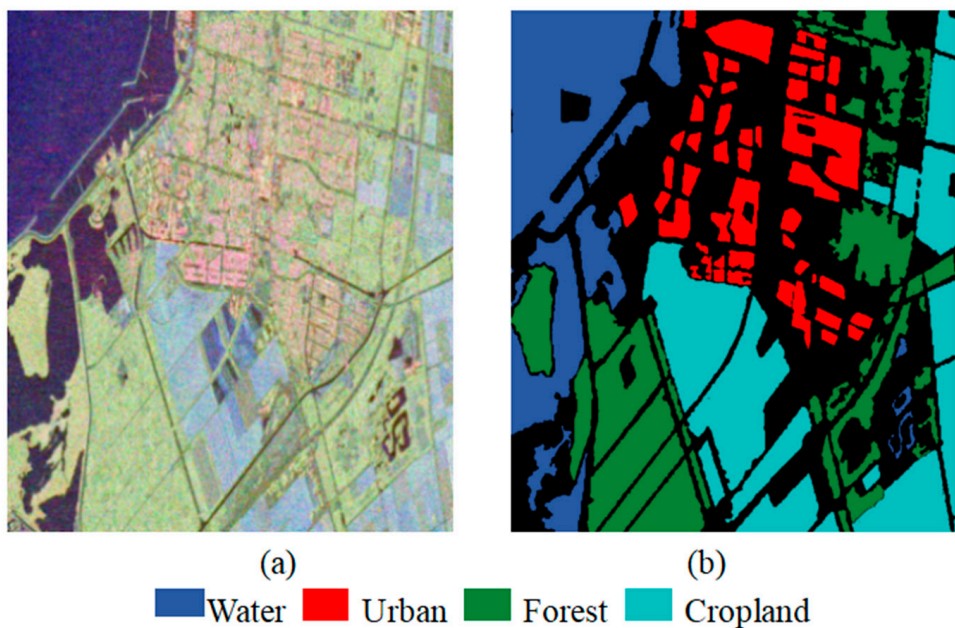

**Figure 6.** Flevoland II dataset: (**a**) Pauli image. (**b**) Ground-truth image.

The third PolSAR dataset is C-band PolSAR data acquired by RADARSAT-2 in San Francisco in April 2008 with $1300 \times 1300$ pixels and a resolution of $10m \times 5m$. It contains 5 terrain objects: high-density, low-density, water, developed, and vegetation, and Figure 7a denotes the Pauli RGB map of this image. Figure 7b shows the appropriate ground-truth map of this image, and each type of crop is identified by one color.

The fourth dataset is a C-band PolSAR data of China, Shaanxi, Xi'an, acquired by RADARSAT-2 in 2010, which has $512 \times 512$ pixels. The resolution of this image is $8m \times 8m$. It contains three different categories: bench land, urban, and river. The Pauli RGB map of the image is shown in Figure 8a, and the corresponding ground-truth map of this image is shown in Figure 8b.

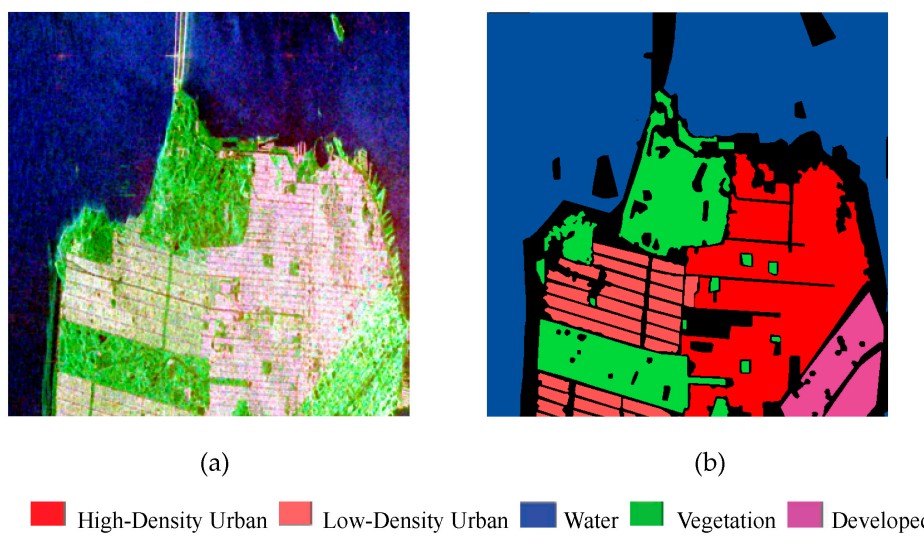

| | High-Density Urban | | Low-Density Urban | | Water | | Vegetation | | Developed |

**Figure 7.** RADARSAT-2 data in San Francisco: (**a**) Pauli image. (**b**) Ground-truth image.

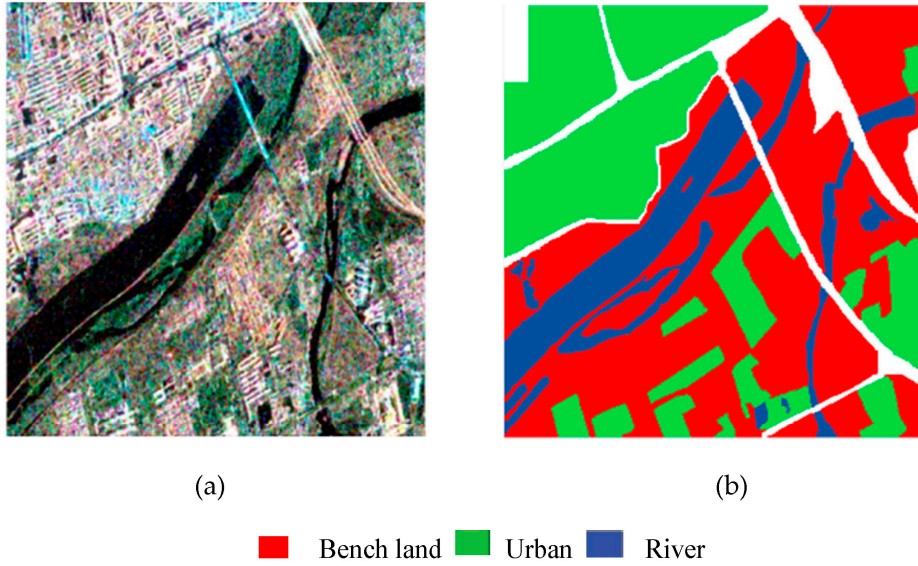

| | Bench land | | Urban | | River |

**Figure 8.** RADARSAT-2 data in Xi'an: (**a**) Pauli image. (**b**) Ground-truth image.

*3.2. Experimental Design Discussion of Key Parameters*

In this subsection, the Flevoland I dataset is utilized as an example to examine the three key parameters of our proposed method: the window size of samples between their neighborhood pixels, the number of support samples, and the batch size of training process.

### 3.2.1. Influence of Sample Window Size on Proposed Method

In our proposed method, we take the whole neighborhood pixel of a center pixel as the input data. Therefore, the window size $w \times w$ of the neighborhood pixels is an important parameter which affects the classification accuracy of this approach. In order to only analyze the effect of window size, we fix the other parameters. The number of support samples for each category is set to 10 and the batch size is set to 75. In this part, the range of $w \times w$ is set from $5 \times 5$ to $17 \times 17$.

Figure 9 shows the OA of the Flevoland I dataset for the proposed method with different window sizes. From Figure 9, it has been observed that OA gradually grows with increasing window size in the beginning. This is primarily due to the fact that when the window is relatively small, as the window size $w \times w$ increases, it contains more neighborhood pixels, provides more spatial information, and is more helpful for the

classification of the center pixels. However, when the window reaches a certain range, some pixels in the neighborhood are farther away from the center pixel. The addition of this neighborhood pixel information will affect the decision of the center pixel category and thus will affect the classification accuracy. For the aforementioned reason, the window size is set to $15 \times 15$ throughout the following experiments.

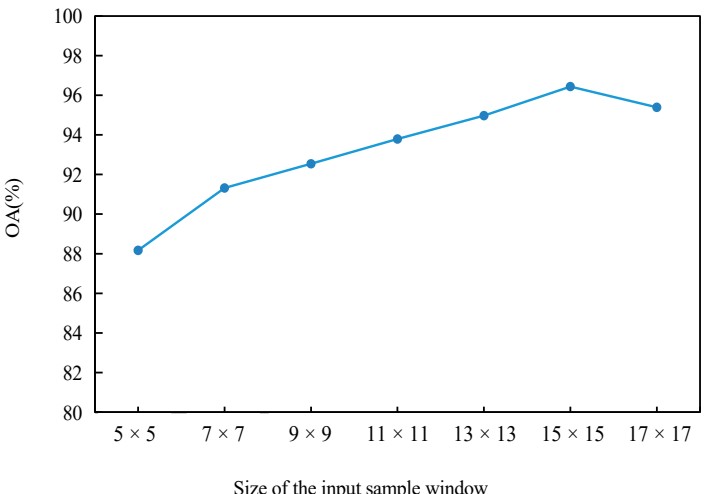

**Figure 9.** Size of the input sample window and OA.

### 3.2.2. Influence of Labeled Samples on Proposed Method

In this section, we only consider the impact of the number of labeled samples of each class on the experimental method. The number of labeled samples is important for the relation network, whose value $K$ will affect the classification results of our proposed. In order to analyze the effect of different numbers of labeled samples and to select the optimal value for our proposed method, we test the classification performance of $K$ support samples of each class from 1 to 10.

Figure 10 represents the classification accuracy of the proposed method with varying numbers of labeled samples on the Flevoland I dataset. From Figure 10, we can observe that the highest OA of the proposed method is 96.24% when $K = 7$ and is larger than other values. In addition, it can also find that when $K$ is larger than 1, OA is larger than $K = 1$. This shows that the $K$-shot classification results are better than the one-shot classification results for the proposed method. Moreover, in Figure 10, it can be found that the extracted features will be diluted or averaged, and the recognition ability will be reduced when the value of $K$ is too large. Therefore, in the following experiment, the value of $K$ is set to 7.

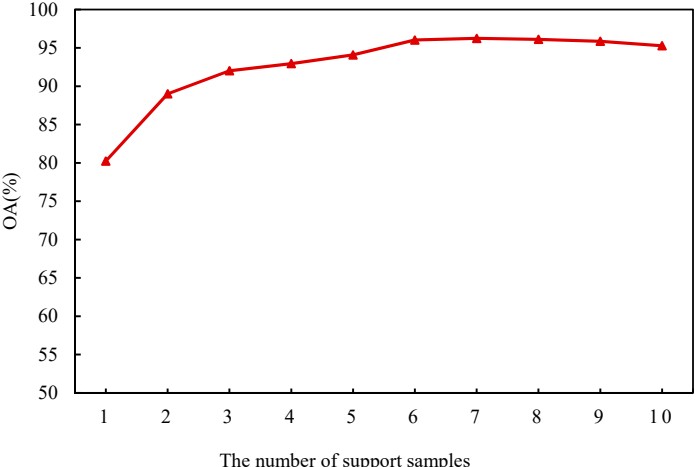

**Figure 10.** Overall accuracy (OA) with different labeled samples.

3.2.3. Influence of Batch Size on Proposed Method

In this section, we only consider the effect of the batch size on classification accuracy. Batch size is a parameter that has a great influence on network training. In theory, if the batch size is small, the difference between adjacent mini-batches is large, causing serious loss curve oscillation and slowing down network convergence. If the batch size is too large, the difference between adjacent mini-batches is small, the loss curve is smooth, and the convergence speed is relatively fast. However, due to the smooth gradient descent, the training only moves in one direction and is prone to falling into the local minimum value. Therefore, in order to choose the appropriate batch size, we use batch sizes ranging from 15 to 150 to train the network and record the classification results of the proposed method with different batch sizes.

Figure 11 depicts the OA of our proposed method for different batch sizes on the Flevoland I dataset. From Figure 11, it can be found that when the batch size is 15, the classification accuracy of the proposed method is the lowest. When the value of the batch size increases gradually, the value of OA also increases gradually and when the value of the batch size is larger than 75, the value of OA tends to be stable. Moreover, in our proposed method, the batch size is usually a multiple of the number of categories. Therefore, on the basis of the above analysis, the batch size is set to $75/15 =$ five times the category $M$ throughout the following experiments.

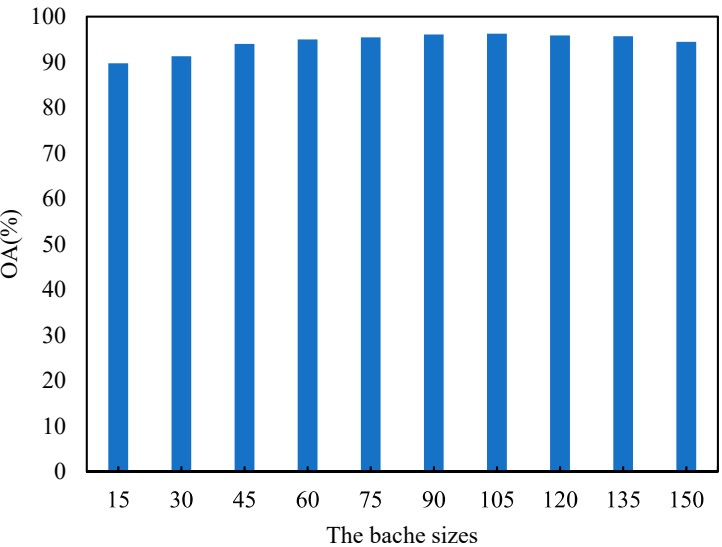

**Figure 11.** The classification accuracy varies with batch size.

## 4. Experimental Results

The four above-mentioned real PolSAR datasets are used to validate the effectiveness of the approach proposed in this section. For these datasets, the size of the input sample window is $15 \times 15$. The number of labeled samples is seven and the batch size is set to five times the number of categories $M$. The overall accuracy (OA) and classification result maps are used to evaluate the property of the proposed method. In order to evaluate the function of each part, the proposed method (3D-ARRN) was compared with five methods: RN, RN-SWANet, RRN, RRN-SWANet, and 3D-RRN. Where RN-SWANet represents the RN combined with SWANet, RRN represents the RN combined with the residual structure, RRN-SWANet represents the RRN combined with SWANet, and 3D-RRN represents the RRN combined with 3D-CNN and residual structure. All these comparison methods use almost similar parameter settings, including the RN structure, SWANet, 3D-CNN structure, residual structure, network layers, the size of filter, and the size of input patches.

### 4.1. Classification Results of the Flevoland I Dataset

Figure 12 depicts the classification results, while Table 3 lists the classification accuracies. The best results are shown in bold. As depicted in Table 3, the OA of the RN, RN-SWANet, RRN, RRN-SWANet, 3D-RRN, and 3D-ARRN are 87.05%, 89.70%, 92.57%, 94.22%, 95.23%, and 96.22%, respectively. Obviously, the proposed method achieved the highest OA among these five comparison methods, which is 9.17% higher than the RN model under the same condition. This result shows that the proposed network structure can significantly enhance accuracy of the RN network in PolSAR image classification. Compared to the RN-SWANet and RN, the OA of RN-SWANet is 2.65% higher than the RN model. This shows that the proposed SWANet module with feature selection in an RN can significantly enhance the classification accuracy of the RN method. Similarly, through comparing the RRN-SWANet, RRN, and RN, it can be found that by introducing the residual structure, the depth of the network is increased and the obtained feature information is more helpful to image classification. In comparing 3D-RRN and RRN, the OA of 3D-RRN is 1.01% higher than RRN. This shows that 3D-CNN increases the mutual relations between different channel features and effectively improves the classification accuracy of the network. Through these comparative experiments, it can be found that each module of the proposed network structure plays the positive effect in the final classification performance. These comparative results also show the effectiveness of our proposed method.

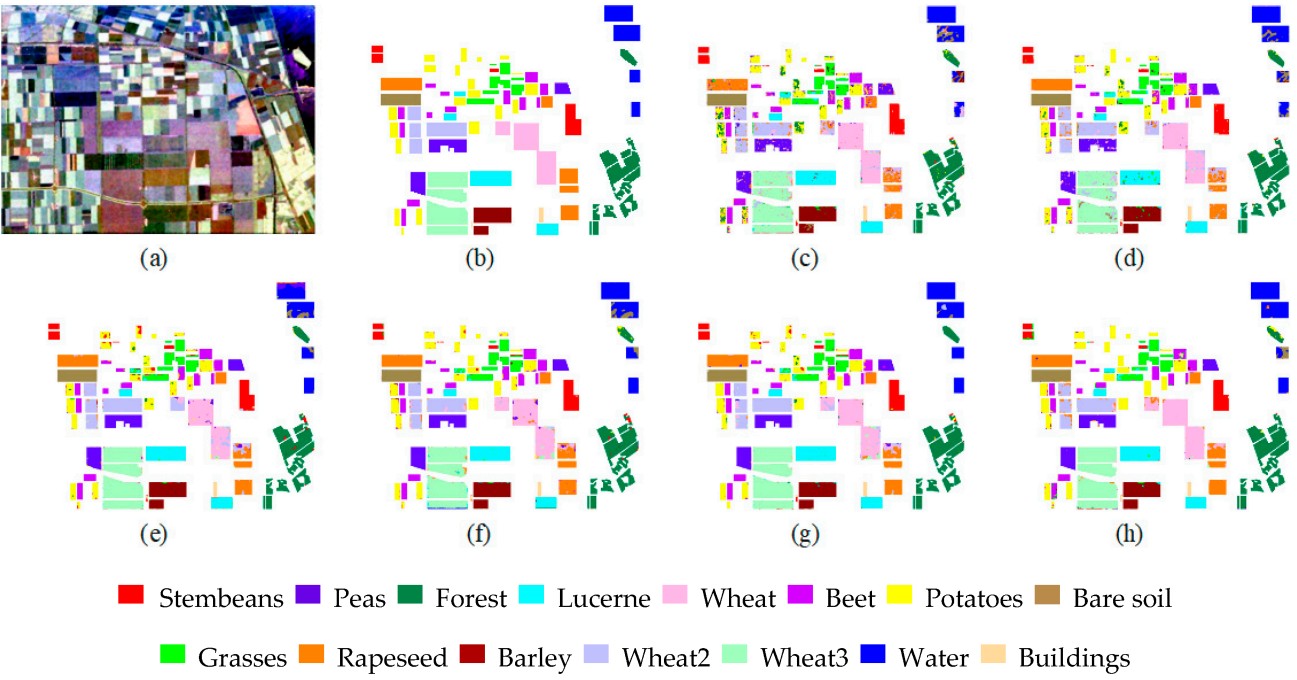

**Figure 12.** Classification results in the Flevoland I area. (**a**) Pauli image. (**b**) Ground-truth image. (**c**) RN. (**d**) RN-SWANet. (**e**) RRN. (**f**) RRN-SWANet. (**g**) 3D-RRN. (**h**) 3D-ARRN.

**Table 3.** The classification accuracy (%) of different methods with 7 labeled samples per class on Flevoland I.

| Method / Region | RN | RN-SWANet | RRN | RRN-SWANet | 3D-RRN | 3D-ARRN |
|---|---|---|---|---|---|---|
| Stem beans | 97.19 | 96.05 | 97.27 | 98.10 | 96.73 | 96.54 |
| Rapeseed | 79.25 | 86.14 | 90.24 | 92.75 | 88.94 | 93.20 |
| Bare soil | 99.71 | 99.98 | 100 | 99.49 | 99.96 | 99.71 |
| Potatoes | 74.64 | 91.15 | 86.02 | 95.27 | 95.81 | 99.16 |
| Beet | 88.38 | 89.42 | 95.21 | 97.77 | 93.12 | 93.76 |

**Table 3.** *Cont.*

| Method / Region | RN | RN-SWANet | RRN | RRN-SWANet | 3D-RRN | 3D-ARRN |
|---|---|---|---|---|---|---|
| Wheat 2 | 84.24 | 88.24 | 96.02 | 92.40 | 95.06 | 91.58 |
| Peas | 91.28 | 92.88 | 98.59 | 99.82 | 99.70 | 98.68 |
| Wheat 3 | 90.92 | 88.82 | 97.58 | 94.78 | 98.43 | 98.86 |
| Lucerne | 97.39 | 93.61 | 96.35 | 94.04 | 96.94 | 98.24 |
| Barley | 93.57 | 96.28 | 96.63 | 97.51 | 99.14 | 98.22 |
| Wheat | 81.83 | 77.32 | 81.96 | 88.74 | 86.73 | 97.07 |
| Grasses | 87.72 | 91.09 | 91.36 | 95.02 | 94.58 | 92.00 |
| Forest | 93.87 | 96.20 | 96.16 | 94.95 | 97.12 | 97.71 |
| Water | 77.52 | 83.97 | 83.83 | 87.91 | 96.52 | 90.45 |
| Building | 88.60 | 91.07 | 93.54 | 97.66 | 95.74 | 98.35 |
| OA | 87.05 | 89.70 | 92.57 | 94.22 | 95.23 | 96.22 |

Moreover, compared with Figure 12c–h, the classification map of Figure 12h is obviously superior to Figure 12c–g. This result also proves that our proposed method can achieve effective classification with few labeled samples.

### 4.2. Classification Results of the Flevoland II Dataset

For this dataset, each category randomly selects seven labeled samples as support samples. Figure 13c–h depicts the classification results and Table 4 shows the classification accuracies.

As shown in Table 4, 3D-ARRN achieved the best accuracy at 95.03% among the comparison methods, and it exhibited significant performance improvements in the majority of categories, such as urban, water, and cropland area. In comparing 3D-ARRN with RN, RN-SWANet, RRN, RRN-SWANet, and 3D-RRN, 3D-ARRN obtained the highest OA. Through the comparison between these methods, it can also be discovered that each module of our proposed method contributes significantly to enhancing the classification accuracy of PolSAR images.

**Table 4.** The classification accuracy of different methods with 7 labeled samples per class on Flevoland II.

| Method / Region | RN | RN-SWANet | RRN | RRN-SWANet | 3D-RRN | 3D-ARRN |
|---|---|---|---|---|---|---|
| Urban | 88.22 | 95.03 | 93.17 | 89.01 | 95.46 | 96.19 |
| Water | 99.21 | 98.11 | 98.65 | 98.56 | 97.50 | 98.86 |
| Forest | 92.78 | 91.69 | 94.04 | 92.26 | 91.09 | 92.78 |
| Cropland | 80.23 | 84.16 | 84.13 | 89.95 | 92.41 | 93.20 |
| OA | 89.98 | 91.55 | 92.08 | 92.79 | 93.84 | 95.03 |

Compared with Figure 13c–h, the classification result shown in Figure 13h is closest to the ground truth. As depicted in Figure 13c, many regions are misclassified. In Figure 13d,e, there are a lot of isolated points in homogeneous regions, which are misclassification points. In Figure 13f,g, these isolated points have been greatly reduced, particularly in the urban and cropland categories. Furthermore, isolated points in Figure 13h are significantly fewer than in Figure 13c–g. This result also indicates the validity of the proposed method again and demonstrates that the method can achieve effective classification with few labeled samples.

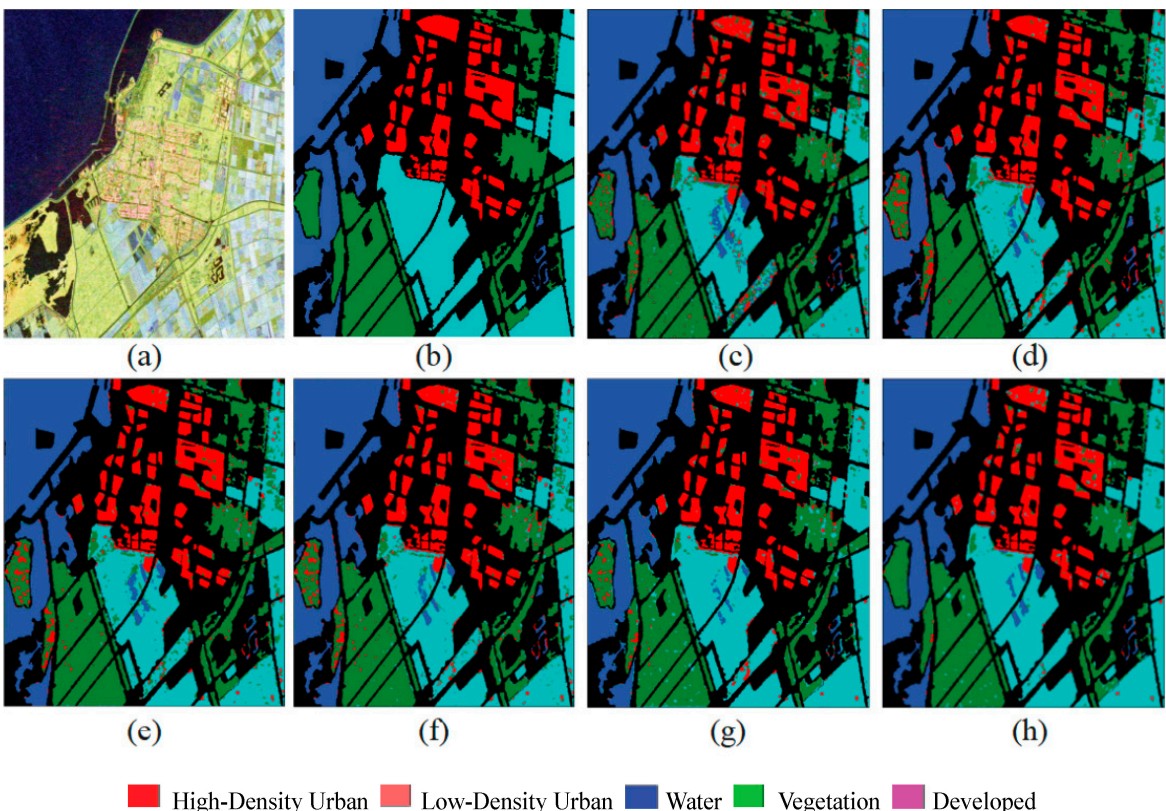

<span style="color:red">■</span> High-Density Urban <span style="color:salmon">■</span> Low-Density Urban <span style="color:blue">■</span> Water <span style="color:green">■</span> Vegetation <span style="color:magenta">■</span> Developed

**Figure 13.** Classification results in the Flevoland II area. (**a**) Pauli image. (**b**) Ground-truth image. (**c**) RN. (**d**) RN-SWANet. (**e**) RRN. (**f**) RRN-SWANet. (**g**) 3D-RRN. (**h**) 3D-ARRN.

### 4.3. Classification Results of the San Francisco Datasets

We randomly picked seven labeled samples for each category as support samples in this dataset. It contains five different terrain types, which means that a total of 35 labeled samples were picked as training samples. The visual classification results are depicted in Figure 14. Table 5 shows the classification accuracy corresponding to each category.

As depicted in Table 5, the classification accuracy of the RN, RN-SWANet, RN-Res, RRN-SWANet, 3D-RRN, and 3D-ARRN are 89.62%, 92.22%, 93.78%, 94.30%, 94.39%, and 96.09%, respectively. This indicates that 3D-ARRN obtained the highest classification accuracy in these comparison methods. These comparisons show that all parts of the suggested method can significantly improve the classification accuracy. From the final classification results, we can observe the ability of our proposed method to solve few-sample problems.

Figure 14 indicates the classification result maps of different methods on the San Francisco area. Compared with Figure 14c–h, our proposed classification map of Figure 14h is better than other methods. As is shown in Figure 14, some water areas of Figure 14c–e are mistaken for the type of vegetation area. In Figure 14f,g, these isolated points are greatly reduced, especially in vegetation, high-density urban, and water areas. In Figure 14h, the classification results of water areas, high-density cities, and developed areas are significantly better than those of other methods in these three areas. The experimental results also demonstrate that the suggested method is efficient.

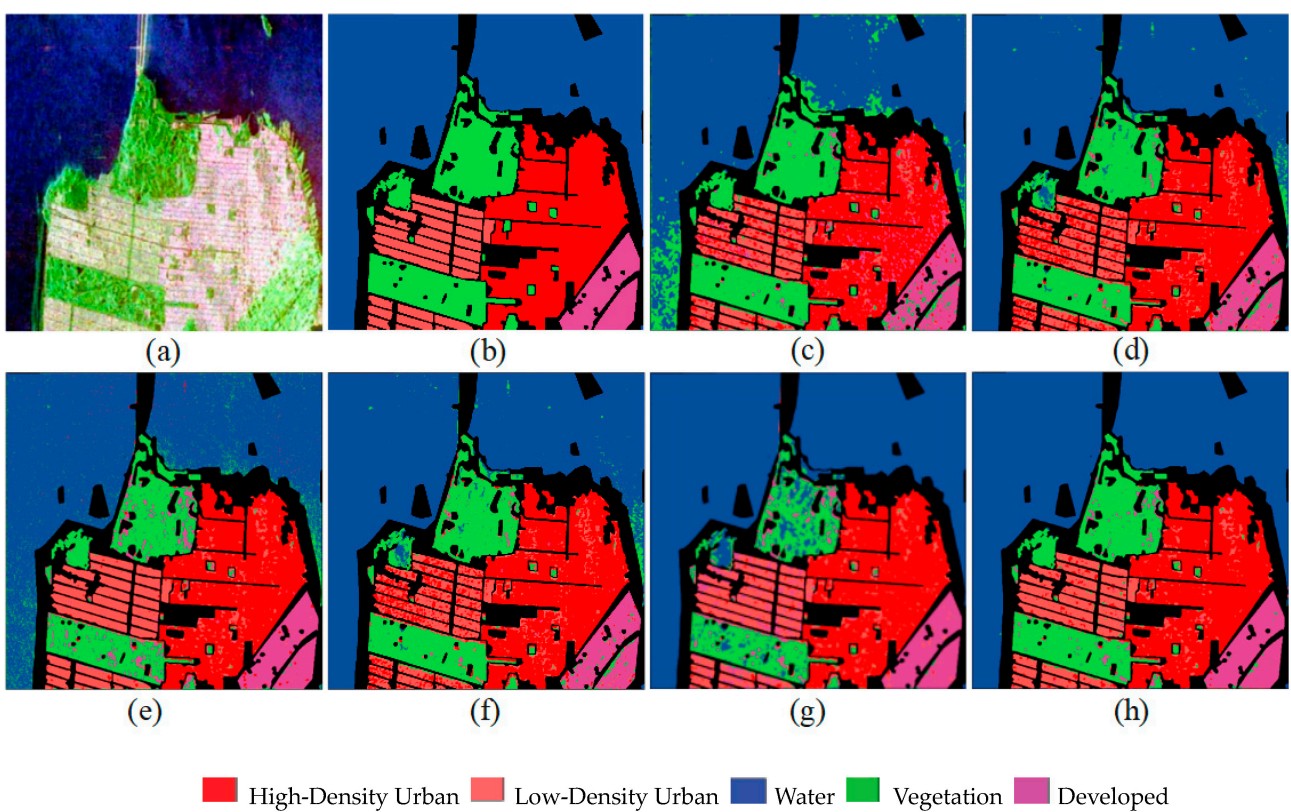

| | High-Density Urban | Low-Density Urban | Water | Vegetation | Developed |

**Figure 14.** Classification results of the San Francisco dataset. (**a**) Pauli image. (**b**) Ground-truth image. (**c**) RN. (**d**) RN-SWANet. (**e**) RRN. (**f**) RRN-SWANet. (**g**) 3D-RRN. (**h**) 3D-ARRN.

**Table 5.** The classification accuracy of all methods with 7 labeled samples per class on the San Francisco dataset.

| Method / Region | RN | RN-SWANet | RRN | RRN-SWANet | 3D-RRN | 3D-ARRN |
|---|---|---|---|---|---|---|
| Water | 93.07 | 98.80 | 99.79 | 99.43 | 99.75 | 99.98 |
| Vegetation | 94.14 | 87.50 | 75.33 | 72.94 | 83.37 | 90.28 |
| Low-Density Urban | 73.67 | 67.43 | 95.52 | 93.48 | 79.99 | 90.00 |
| High-Density Urban | 84.86 | 89.97 | 90.78 | 96.61 | 95.15 | 92.73 |
| Developed | 87.17 | 89.21 | 96.05 | 96.92 | 92.99 | 97.53 |
| OA | 89.62 | 92.22 | 93.78 | 94.30 | 94.39 | 96.09 |

### 4.4. Classification Results of the Xi'an Datasets

The dataset consists of three main categories: bench land, urban, and river. We randomly selected seven labeled samples from each class as the support set, which means that a total of 21 labeled samples were selected as training samples. The classification results are shown in Figure 15a–h and the classification accuracy is listed in Table 6.

As shown in Table 6, in comparing 3D-ARRN with RN, RN-SWANet, RRN, RRN-SWANet, and 3D-RRN methods, it can be found that the OA and Kappa of 3D-ARRN are higher than the other comparison methods. This is because when the number of labeled samples is few, these network models cannot be fully trained, and the fitting capability is poor. Moreover, these comparisons indicate that each part of the proposed method can effectively improve its classification accuracy. This also indicates the ability of the proposed method to solve few-sample problems and proves the superiority of the proposed method.

Figure 15 shows the classification result maps of different methods in the Xi'an area. Compared with Figure 15a–h, the proposed classification map of Figure 15h is better than the other methods. In Figure 15c–h, the isolated points are greatly reduced, especially in

bench land and urban areas. As shown in Figure 15h, the proposed method has significant advantages over the other methods in bench land and urban areas. These experiment results also prove the effectiveness of the proposed method.

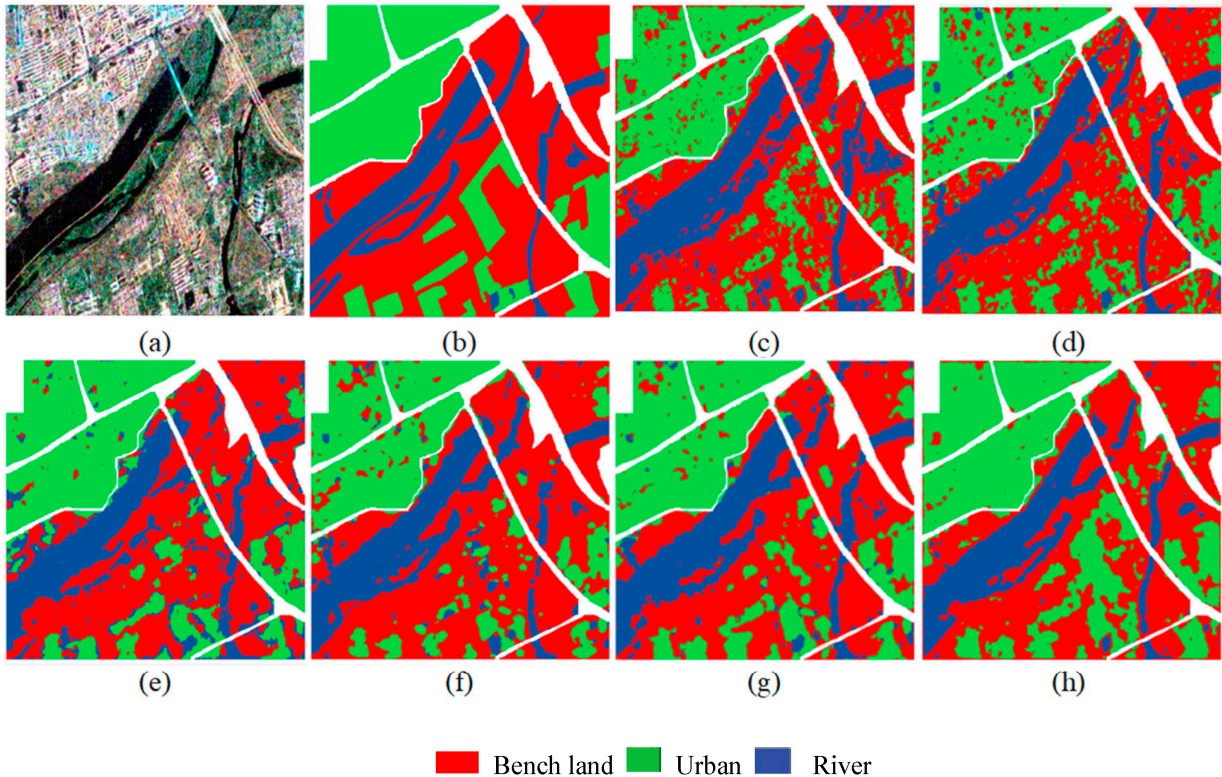

Bench land    Urban    River

**Figure 15.** Classification results of the Xi'an dataset. (**a**) Pauli image. (**b**) Ground-truth image. (**c**) RN. (**d**) RN-SWANet. (**e**) RRN. (**f**) RRN-SWANet. (**g**) 3D-RRN. (**h**) 3D-ARRN.

**Table 6.** The classification accuracy (%) of different methods with 7 labeled samples per class on Xi'an dataset.

| Method / Region | RN | RN-SWANet | RRN | RRN-SWANet | 3D-RRN | 3D-ARRN |
|---|---|---|---|---|---|---|
| Bench land | 77.16 | 80.82 | 80.69 | 82.26 | 82.36 | 84.19 |
| Urban | 79.66 | 76.92 | 84.93 | 82.43 | 85.08 | 92.41 |
| River | 93.94 | 94.14 | 82.51 | 91.26 | 93.04 | 88.97 |
| OA | 80.56 | 81.44 | 82.86 | 84.24 | 85.27 | 88.12 |

### 4.5. Comparisons with Other PolSAR Classification Methods

To verify the efficiency of our suggested method, we compared it to other recent PolSAR classification methods. Table 7 shows the classification accuracy of these methods on the first dataset: the Flevoland I area. In the 3D-ARRN, SVM [15], Wishart [13], 3D-CNN [42], SF-CNN [26], CNN-PL, and 3D-CNN + PL classification methods, seven labeled samples from each class were chosen at random as training samples. CNN + PL represents the CNN combined with the pseudo-label generated by the superpixel algorithm. 3D-CNN + PL represents the CNN combined with the pseudo-label generated by the superpixel algorithm. In the STLLE [44], MCCNN [45], MAE [46], and CV-3D-CNN [47] classification methods, 1% of the labeled samples are used as the training samples for the other four methods.

As can be observed from Table 7, the OA of the 3D-ARRN proposed is much higher than that of the SVM, Wishart, 3D-CNN, SF-CNN, CNN + PL, and 3D-CNN + PL methods. When there are seven labeled samples in each class, the accuracy rate of the proposed

method is higher than that of the comparison method in most categories. Moreover, in comparing the proposed method with the STLLE, MCCNN, MAE, and CV-3D-CNN methods, the OA of the proposed method with 105 training samples is still higher than that of the STLLE, MCCNN, MAE, and CV-3D-CNN methods with 1682 training samples. These comparisons demonstrate that our proposed method can effectively solve the problem of limited samples and that it is significantly better than the other methods under the same conditions.

**Table 7.** The classification accuracy (%) of various methods on the Flevoland I dataset.

| Method / Region | 3D-ARRN | SVM | Wishart | 3D-CNN | SF-CNN | CNN +PL | 3D-CNN +PL | STLLE | MC-CNN | MAE | CV-3D-CNN |
|---|---|---|---|---|---|---|---|---|---|---|---|
| Sample Number | Labeled Sample Per Class (The Total Number Is 105) 7 | | | | | | | Training Ratio (The Total Number Is 1682) 1% | | | |
| Stem beans | 96.54 | 66.23 | 89.72 | 81.11 | 87.52 | 99.16 | 98.28 | 97.35 | 97.01 | 95.91 | 98.63 |
| Rapeseed | 93.20 | 56.91 | 71.13 | 11.42 | 91.26 | 91.22 | 86.47 | 86.87 | 91.94 | 84.11 | 97.48 |
| Bare soil | 99.71 | 79.51 | 98.04 | 94.03 | 100 | 100 | 99.86 | 97.46 | 91.70 | 92.62 | 92.74 |
| Potatoes | 99.16 | 48.05 | 74.99 | 64.92 | 93.53 | 85.89 | 91.51 | 93.35 | 96.04 | 89.64 | 93.60 |
| Beet | 93.76 | 68.66 | 91.32 | 85.25 | 96.25 | 96.82 | 96.95 | 96.99 | 93.26 | 95.77 | 95.21 |
| Wheat 2 | 91.58 | 31.63 | 64.17 | 73.13 | 89.84 | 88.29 | 92.81 | 83.85 | 98.14 | 81.02 | 95.73 |
| Peas | 98.68 | 70.41 | 94.64 | 38.38 | 97.78 | 95.25 | 98.33 | 97.61 | 97.76 | 96.42 | 87.65 |
| Wheat 3 | 98.86 | 59.97 | 76.77 | 69.91 | 84.84 | 96.98 | 97.32 | 95.06 | 97.60 | 95.06 | 99.44 |
| Lucerne | 98.24 | 65.57 | 93.69 | 81.22 | 94.79 | 94.37 | 90.18 | 94.65 | 97.77 | 95.34 | 84.81 |
| Barley | 98.22 | 55.10 | 91.89 | 82.14 | 94.18 | 97.95 | 94.39 | 89.37 | 99.39 | 95.98 | 84.14 |
| Wheat | 97.07 | 46.85 | 79.85 | 53.02 | 92.54 | 89.33 | 83.35 | 88.32 | 86.24 | 91.57 | 98.79 |
| Grasses | 92.00 | 61.21 | 64.72 | 55.68 | 84.61 | 88.43 | 95.29 | 81.68 | 97.82 | 86.41 | 72.39 |
| Forest | 97.71 | 82.51 | 51.10 | 58.17 | 95.24 | 96.77 | 95.83 | 90.19 | 99.14 | 91.13 | 99.85 |
| Water | 90.45 | 69.49 | 81.31 | 74.42 | 90.80 | 83.81 | 92.89 | 98.87 | 98.15 | 98.02 | 99.95 |
| Building | 98.35 | 61.77 | 83.94 | 58.77 | 88.71 | 98.08 | 98.63 | 86.81 | 98.38 | 84.09 | 96.22 |
| OA | 96.22 | 61.59 | 80.49 | 65.44 | 92.13 | 92.63 | 93.14 | 92.33 | 95.83 | 92.01 | 93.42 |

## 5. Conclusions

In this study, a new 3D-ARRN network was proposed for PolSAR image classification when labeled samples are few. Our method avoids the cost of manually labeling a large number of samples and effectively improves the classification accuracy under few-samples. The experiment results on four typical datasets from different radar systems indicate that the proposed method was superior to the comparative classification method in these four tested cases. The advantages of our proposed method have been demonstrated by these experiments. That is, (1) This method can solve the classification problem of PolSAR images with few samples effectively and it can obtain better classification results when there are only a few labeled samples in each category; (2) the training samples construction method based on characteristics of PolSAR data provides a new idea for PolSAR terrain classification with a limited number of samples and can also be applied to the PolSAR image classification method based on meta-learning; (3) the proposed attention mechanism of a SWANet in this paper can effectively play the role of feature selection and improve classification accuracy. Compared with other RN-based methods (RN, RN-SWANet, RRN, RRN-SWANet, and 3D-RRN), our proposed method obtained the higher classification accuracy under the same conditions, which shows that each part of an RN model can effectively improve the proposed method. Moreover, compared with several typical PolSAR image classification methods, our proposed method has significant advantages, especially when labeled samples are few. In the future, we will focus on the physical scattering mechanism and study PolSAR classification with few samples driven by the physical scattering mechanism.

**Author Contributions:** W.H. implemented the main ideas, carried out the evaluation, and wrote the manuscript. Y.Z. and C.Z. provided suggestions on presenting the results and writing the paper. X.J. provided some suggestions on experimental design. All authors have read and agreed to the published version of the manuscript.

**Funding:** This work was supported by the National Natural Science Foundation of China [No. 61901368], the Special scientific research plan project of the Shaanxi Provincial Department of Education [19JK0798], and the Natural science foundation of Shaanxi Province [2019JQ-377].

**Data Availability Statement:** The original PolSAR data are publicly available and can be found: https://ietr-lab.univ-rennes1.fr/polsarpro-bio/sample_datasets/. The data presented in this study are available on reasonable request from the corresponding author.

**Conflicts of Interest:** The authors declare no conflict of interest.

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
