# Peer review of "PolSAR Image Classification Based on Relation Network with SWANet"

_remotesensing, doi:10.3390/rs15082025_

Round 1

Reviewer 1 Report

Comments to the Author:

This work deals with PolSAR image classification with deep learning models. Generally speaking, the topic is interesting and the presentation is clear. The following major concerns should be fully considered during the revision.

1. Few-shot classification is very important. Please give a definition about the “few-shot” within the scope of PolSAR image classification.

2. Experimental studies should be enhanced. The current version still follows the fashion that trains and tests datasets one by one. Actually, as demonstrated in many publications, such fashion can achieve relatively very high OAs. But, the classifier’s generalization performance is not examined. Please add experimental studies to validate the generalization performance.

3. In section 2.1,why should the author extract roll-invariant polarimetric feature, and what is the difference between them and polarimetric features based on target decomposition?

4. The reviewer would like to point out a couple of points in this sentence to improve its readability. a)  Page 3, Line 104, “focusing the characters of PolSAR image, a new few-shot learning classification method ...” ; b) Page 3, Line 106, However, RN requires source domain data with a good deal of labeled samples to train.  

5. Page 4, Line 143-145, the variables D, E, and F in Eq.2 should be expressed in italics.

6. The variableappears in both Eq.5 and Eq.8, but the meaning of the two expressions should be different. The author should carefully check to distinguish between different variables and use different symbols to represent them.

7. The variable W in Eq. 6 and Eq. 8 represents different meanings. The meaning of the variable V in Eq. 9 to Eq.12 and the variable V in Eq. 6 are also different. Please use different variable symbols to represent them.

Author Response

Thank you very much for your letter and the comments concerning our manuscript entitled. 

Those comments are all valuable and very helpful for revising and improving our paper, as well as the important guiding significance to our research. We have studied the comments carefully and have made correction which we hope meet with approval.

Below we provide our point-by-point responses to the comments. We have reproduced the original comments in italic, followed by our responses. The modifications in the revised paper are marked in red.

The detailed response can be found in the attachment below.

Reviewer 2 Report

This paper proposes an attention-based 3D residual relation network (3D-ARRN) for Few-Shot PolSAR image Classification. The experimental results confirm the performance of this method. However, there are some unclear parts in the manuscript. 1. How to set the value of Ns in Section.2.4? 2. Normally, each training sample should be independent, but the method you use to generate pseudo-labels in the entire PolSAR image using superpixel segmentation cancels the independence between samples and is not universal. 3. In this paper, the author proposes a spatial weighted attention network (SWANET). The Convolutional Block Attention Module (CBAM) is a typical spatial channel attention mechanism that has achieved significant results in many applications. What are the advantages of the proposed SWANET compared to CBAM? 4. Which of the algorithms in your CNN-based comparative experiments are using the superpixel algorithm for pseudo-label generation.

5. Please provide other detailed parameters of the proposed classification network, such as the number of convolution layers, pooling layers, convolution kernel parameters, etc.

6. The reviewer would like to point out that there are several such sentences in the manuscript that can either be omitted since they don’t seem to be contributing any additional information (Ex. Page 2, Line 50-51, Page 5, Line 172-173 ) .

7. Page4, Line 145, The font size of variables is inconsistent in Table 1, please modify to a uniform size.

8. Some grammatical mistakes and improper expression should also be modified.

Author Response

(The authors gave the same response as above.)

Reviewer 3 Report

What are the key characteristics and advantages of PolSAR technology compared to traditional SAR technology in remote sensing applications?

How can machine learning algorithms be used in conjunction with PolSAR data for land cover classification and feature extraction?

What are some common preprocessing techniques used for PolSAR data, and how do they impact subsequent classification performance?

What is the role of feature selection in PolSAR image classification, and what are some commonly used feature selection methods in this context?

How does polarimetric target decomposition work, and what are some of its applications in PolSAR image analysis?

What are some challenges and limitations associated with PolSAR data and their impact on classification accuracy?

How can domain adaptation techniques be used to improve the generalization of PolSAR image classifiers across different geographical regions or time periods?

What is the role of transfer learning in PolSAR image classification, and how can it be used to leverage pre-trained models for improved classification performance?

What are some recent advances in PolSAR image classification, and how do they compare to traditional machine learning-based approaches?

How can PolSAR data and machine learning be used for other remote sensing applications beyond land cover classification, such as object detection or change detection?

Author Response

(The authors gave the same response as above.)

Reviewer 4 Report

In this paper, a polarimetric synthetic aperture radar (PolSAR) images classification method is proposed. By utilizing a multilayer CNN with residual structure to extract deep polarimetric features, and a spatial weighted attention network (SWANet) to concentrate the feature information, the proposed method achieved a higher classification result than other methods with few labeled data. Some specific comments to are listed as follows:

1.         There seems to be some mistakes of the explanation of formulas. For example, the author tries to explain the meaning of formula (3) on line 172, however, xi is explained twice with different meanings, while xj does not appear.

2.         In lines 194 - 196, we’re told that training set has its own separate label space and does not intersect with support set and query set. However, in lines 201-204, it’s said that the support set and query set are randomly select from training set, which indicate that they definitely have intersection in their label spaces. This may cause confusion for readers who are not in this specific field.

3.         The author needs to pay more attention to the format of the paper. For example, there is something wrong about the indentation of the paragraph in section 3.2.

4.         More comparative experiments should be added. According to section 4.1-4.3, the ablation study is implemented in all these 3 datasets, however, comparative experiments only implement based on one of them. Is there any specific reason for this?

5.         The authors tried to conclude the recent advances. However, there are still some relevant works about the application of deep learning technology in SAR. It is recommended to add these works to the reference, such as

https://www.doi.org/10.1109/TGRS.2017.2777868

https://www.doi.org/10.1109/TGRS.2023.3248040

https://www.doi.org/10.1109/LGRS.2019.2923403

Author Response

(The authors gave the same response as above.)

Round 2

Reviewer 4 Report

I have no other questions.